# Multicenter Study of Benign Thyroid Nodules with Radiofrequency Ablation: Results of 762 Cases over 4 Years in Taiwan

**DOI:** 10.3390/jpm12010063

**Published:** 2022-01-06

**Authors:** Wei-Che Lin, Cheng-Kang Wang, Wen-Hung Wang, Chi-Yu Kuo, Pi-Ling Chiang, An-Ni Lin, Jung Hwan Baek, Ming-Hsun Wu, Kai-Lun Cheng

**Affiliations:** 1Department of Diagnostic Radiology, Kaohsiung Chang Gung Memorial Hospital, Chang Gung University College of Medicine, Kaohsiung 833, Taiwan; alex@cgmh.org.tw (W.-C.L.); moonrbt@gmail.com (C.-K.W.); lovage@cgmh.org.tw (P.-L.C.); miracoli1126.AL@gmail.com (A.-N.L.); 2Department of Otolaryngology, Cathay General Hospital, Taipei 106, Taiwan; ent.taiwan@gmail.com; 3Department of Otolaryngology, Sijhih Cathay General Hospital, New Taipei City 221, Taiwan; 4School of Medicine, Fu-Jen Catholic University, New Taipei City 242, Taiwan; 5Department of Surgery, MacKay Memorial Hospital, Taipei 104, Taiwan; kiwiyu@gmail.com; 6Department of Radiology and Research Institute of Radiology, University of Ulsan College of Medicine, Asan Medical Center, Seoul 05505, Korea; radbaek@naver.com; 7Department of Surgery, National Taiwan University Hospital, Taipei 100, Taiwan; 8Department of Medical Imaging, Chung Shan Medical University Hospital, Taichung 402, Taiwan; 9School of Medical Imaging and Radiological Sciences, Chung Shan Medical University, Taichung 402, Taiwan

**Keywords:** goiter, radiofrequency ablation, Taiwan, thyroid gland, ultrasound

## Abstract

Background: To evaluate the effectiveness and safety of radiofrequency ablation (RFA) treatment for benign thyroid nodules (BTNs) in five medical centers in Taiwan. Methods: From April 2016 to July 2020, 762 patients underwent ultrasound guided RFA treatment of 826 benign thyroid nodules at five medical centers in Taiwan. The RFA procedure was performed by radiologists, otolaryngologists, or surgeons. Patients were grouped into three subgroups according to the initial volume of BTNs. The volume reduction ratio (VRR) of each nodule, and complications were analyzed at 1, 3, and 6 months after RFA treatment and the three groups compared. Results: The large nodular group showed greater VRR compared to the other two groups at first 1-month follow-up. At 6-months follow-up, there was no significant difference of VRR among the three groups. Goiters with difference in size can attain a successful VRR (>50%) although different specialists demonstrated variable VRR after 6-months follow-up. A total of 40 (4.8%) complications were reported. All patients recovered spontaneously without surgery intervention. Conclusions: The reliability and safety of RFA for benign thyroid nodules had been established. RFA has gradually become an alternative to surgery in the treatment of benign thyroid nodules in Taiwan.

## 1. Introduction

In 2006, ultrasound (US)-guided percutaneous radiofrequency ablation (RFA) was first reported to treat benign thyroid nodules (BTNs) [1] and has since become an established technique, particularly in Asia and Europe. The RFA electrode needle has an active tip, which heats due to the alternating current passing through it, and is inserted into the nodule under direct ultrasound guidance. The BTN is ablated by application of the trans-isthmic moving shot technique [2], moving the electrode tip backward and forward, from bottom to top. RFA has become a valuable treatment strategy for BTNs due to its non-surgical, minimally invasive characteristics. In addition, thyroid RFA has been demonstrated as an effective treatment [3,4,5] with a relatively low complication rate [6]. Indeed, BTNs have shown a post-ablation volume reduction rate (VRR) from 50% to 80% [3,7,8].

Thyroid RFA using the moving shot technique needle approach has been applied in Taiwan since 2012 [9], and by 2016 had become an established method in three major medical centers including Kaohsiung Chang Gung Memorial Hospital, National Taiwan University Hospital, and Chung-Shan Medical University Hospital. To further promote thyroid RFA, the Taiwan Academy of Tumor Ablation (TATA) holds at least one or two lectures annually, and offers hands-on thyroid RFA training programs for their members. The minimally invasive treatment has thus increased popularity, and has gradually evolved into an acceptable alternative therapy for BTNs in at least ten hospitals in Taiwan.

Although countries which have developed and applied the ablation technique for BTNs have noted positive results, the initial outcomes of this procedure in Taiwan as performed at different hospitals and by different specialists—including radiologists, surgeons, and otolaryngologists; and potentially endocrinologist in the near future—remains unclear. The aims of this multicenter study were to evaluate the therapeutic efficacy, consistency, and safety of thyroid RFA for treatment of BTNs of various sizes in different medical centers and specialist departments in Taiwan.

## 2. Materials and Methods

### 2.1. Patients

From April 2016 to July 2020, the data of more than 1500 patients who had undergone RFA of BTNs were collected from five medical centers in Taiwan (Figure 1 and Appendix A). All patients had nodule-compressed symptoms, cosmetic problems, or were identified incidentally. Patients visited otolaryngologists, surgeons, or radiologists for assessment of thyroid nodules and the nodular composition by ultrasound or CT/MRI. In accordance with a previous study, nodules were classified as ‘solid’, ‘predominant solid’, ‘predominant cystic’, or ‘cyst’ [10]. Core needle biopsy or ultrasound guided fine-needle aspiration cytology (FNAC) was performed before RFA to verify the benign nature of the nodules. A single benign cytological result with ultrasound benign appearance or two benign cytological results was considered at low risk of malignancy [11,12].

Patients refused operations over concerns of complications, scarring, hypothyroidism, or voice change, although they had no contraindication for operation. The exclusion criteria were patients who did not return at each of the follow-up sessions (1-month, 3-month, and 6-month), patients with cyst or predominant cystic nodule, patients with pathological results such as malignancy, patients who received RFA for more than one nodule concurrently, and less than 6 months after receiving RFA. A total of 762 patients presenting with 826 solid benign nodules who underwent RFA for a single nodule were enrolled in this study (Figure 2). For those patients presenting with more than one nodule, RFA was performed for a single nodule per session. Patients were divided into three groups according to the size of the BTNs as measured by ultrasound. This study is a retrospective study and approved by the institutional review board of the five medical centers involved, and informed consent was obtained from all patients prior to the procedure.

### 2.2. Pre-Ablation Assessment

At every visit, data regarding cosmetic score, symptomatic score, and thyroid function were recorded. The cosmetic score was obtained using the following scale: 0, no visible or palpable mass; 1, not visible but palpable mass; 2, visible when swallowing only; 3, an easily visible mass [12]. The symptomatic score was obtained by patients filling out a questionnaire involving five clinical symptoms: compression, cough, difficulty swallowing, pain, and voice change. We allocated one point for each positive symptom, therefore the symptom scores ranged from 0 to 5. Ultrasound was used to evaluate nodular morphology and composition. The size, three orthogonal diameters of the tumors (the largest diameter and two other perpendicular ones), was also measured by ultrasound. The volume of the tumor was calculated using the following equation: *V* = πabc/6 (*V*: volume; a: the largest diameter; b and c: the other two perpendicular diameters) [12]. To evaluate the relationship between nodule volume and treatment efficacy, patients were categorized into three groups: large: a pre-ablation nodule volume >30 mL; medium: a pre-ablation nodule volume of 11–30 mL; and small: a pre-ablation nodule volume ≤10 mL [13].

### 2.3. Ablation Technique

In different medical centers, the ablation procedure was performed by radiologists, surgeons, and otolaryngologists who were experienced in US-guided procedures or trained by the Taiwan Academy of Tumor Ablation (TATA). All patients were arranged in an outpatient setting. Prior to ablation, a solution of 2% lidocaine hydrochloride was used as local anesthesia at the puncture site and around the thyroid gland. RF ablation was performed with RF generator (VIVA, STARmed and M2004, RF Medical) with an internally cooled electrode (18 gauge, with 5 mm, 7 mm, or 1 cm active tip) based on tumor size, and the status of the surrounding critical structures. Percutaneous ultrasound guidance with trans-isthmic approach, passing through thyroid parenchyma with careful observation of the vessels along the approach route was used. An electrode was placed into the thyroid nodule at the deepest portion of the nodule, followed by ablation using the moving shot technique. When all visual fields of the nodule had changed to transient hyperechoic zones, ablation termination was determined.

### 2.4. Follow-Up Evaluation

To evaluate the VRR, 1-month, 3-month, and 6-month post-ablation follow-up ultrasound was performed. The cosmetic score and symptomatic score were also evaluated at time of follow-up. The VRR was assessed by US imaging and was calculated by the following equation: volume reduction ratio (%) = initial volume (mL)–final volume (mL) × 100/initial volume. Major and minor complications were assessed according to the standard terminology of the Society of Interventional Radiology (SIR) [14]. Major complications include SIR classifications C–F; C: requires therapy, minor hospitalization (<48 h); D: requires major therapy, prolonged hospitalization (>48 h); E: permanent adverse sequelae; F: death. Minor complications include SIR classification A–B; A: no therapy, no consequence; B: nominal therapy, no consequence; includes overnight admission for observation only.

### 2.5. Analysis and Statistics

Statistical analysis was performed by using SPSS, version 22 (SPSS, Inc., Chicago, IL, USA). All data were given as the mean ± standard deviation (SD). Demographic characteristics and ultrasound results among the three groups were compared. Standard Chi-square and Fisher’s exact test were used for group comparisons of the categorical data. Group comparisons for the continuous variables data were performed by using the Mann–Whitney U Test and Kruskal–Wallis Test (SPSS, Inc., Chicago, IL, USA). A generalized estimation equation for repeated measure analysis was used to examine the relationship of nodule volume and VRR among groups at each follow-up time. Point *p* value < 0.05 was considered significant.

## 3. Results

### 3.1. Demographic Characteristics

A summary of demographic data at each of the five medical centers is demonstrated in the Table 1 and Appendix A. The number of patients per center ranged from 62 to 376 (total, 762 patients, including 625 female and 137 male patients; mean age, 45.2 years) and the number of nodules per center ranged from 67 to 435 (total, 826 nodules). All enrolled patients had a single ablated thyroid nodule at each session. All ablated nodules were evaluated by ultrasound and defined as a solid or predominant solid component. Of 826 nodules, 180 nodules (21.8%) constituted the large nodular group; 295 nodules (35.7%) constituted the medium nodular group; and the remaining 351 nodules (42.5%) constituted the small nodular group. There were no differences in age or sex distribution, and serum T4 levels between the three groups. Serum T3 and TSH levels showed differences (T3, *p* = 0.045; TSH, *p* = 0.015) between the three groups; however, both were within normal range.

### 3.2. Volume Reduction Ratio (VRR)

The baseline mean nodular volume and respective changes of VRR at each follow-up point after RFA are shown in Table 2 and Figure 3. Prior to RFA, the overall mean baseline volume of the BTNs was 21.51 ± 27.13 mL. After ablation, the 1-month, 3-month, and 6-month mean volumes were 11.71 ± 13.75 mL, 8.07 ± 10.23 mL, and 5.94 ± 8.83 mL, respectively. The results demonstrated that the overall nodular volume reduced significantly after RFA treatment over time (time effect, *p* < 0.001). The volume in each nodular volume group also showed significant reduction over time (time effect, *p* < 0.001). At the 1-month follow-up, the VRR showed a significant reduction in the large nodules group (47.13% ± 21.51), as compared to the small nodules group (30.25% ± 70.10, *p* < 0.001) and medium nodules group (40.85% ± 20.88, *p* = 0.01); while the medium nodular VRR was significantly higher than the small nodular VRR (*p* = 0.035). Notably, at the 6-month follow-up, there were no significant differences of VRR among the three groups (large, 72.51% ± 18.52; medium, 72.12% ± 18.23; small, 74.62% ± 28.64, *p* = 0.15).

### 3.3. Cosmetic and Symptomatic Scores

The cosmetic and symptomatic scores are shown in Table 2. Prior to ablation, there were 678 (88.9%) patients with cosmetic problems, defined as a cosmetic score >0. The mean cosmetic score was elevated in the larger nodular groups before RFA, with the highest score in large group (*p* < 0.001). At the 6-month follow up, the overall cosmetic score had improved from 2.53 ± 1.09 to 0.91 ± 0.97 (*p* < 0.001), with each group achieving significant improvements (*p* < 0.001). The cosmetic score at the 6-month follow-up remained significantly higher in the large nodular group than in the medium nodular group (*p*< 0.001) and small nodular group (*p* < 0.001).

Prior to ablation, there were 584 (76.6%) patients with nodule-related symptoms, defined as a symptoms score >0. The mean symptomatic score of the large nodular group was significantly higher than that of the small nodular group (*p* < 0.001), but similar to the medium nodular group (*p* = 0.21). The overall symptomatic score improved from 1.75 ± 1.41 to 0.13 ± 0.42 at the 6-month follow-up (*p* < 0.001), with each group achieving significant improvement (*p* < 0.001). At the 6-month follow-up, the large nodular group presented higher symptomatic scores than the medium and small nodular groups (*p* < 0.001), however the symptomatic scores of the large nodular group indeed showed significant improvement over baseline (*p* < 0.001).

### 3.4. Complications

A total of 40 (4.8%) complications were reported, including immediate and delayed, as shown in Table 2. Voice changes were noted in 22 patients (2.6%), delayed nodule rupture in 9 patients (1.0%), and ptosis in 2 patients (0.2%). Most patients with voice change recovered within 1–3 months after ablation. Other complications included skin burn in one patient (0.1%), transient hyperthyroidism in two patients (0.2%), and hematoma in four patients (0.4%). The overall complication rate showed significant difference among the three groups. The large nodular group presented a higher complication rate (40 patients: large group, *n* = 21, 11.6%; medium group, *n* = 6, 2.0%; small group, *n* = 13, 3.7%, *p* < 0.001).

## 4. Discussion

This retrospective multicenter study reveals a positive treatment outcome of BTNs with US-guided RFA in Taiwan. All nodules except one were ablated in a single session, and all nodules were classified as solid or predominant solid. As to the primary outcome, our results confirm that the overall VRR was 73.24 ± 23.19% at the 6-month follow-up. Similarly, a previous study reported a VRR range of 52.1–86.1% at 6-months post-ablation [1]. Regarding cosmetic and symptomatic outcomes, a significant improvement (*p* < 0.001) in cosmetic concerns and compressive symptoms (difficulty swallowing, pressure in the neck, and other cosmetic complaints) was demonstrated in all patients during the follow-up period. The findings of this study indicate that RFA is an effective, reliable, and safe treatment for BTNs.

While the effects of RFA treatment are notable, volume reduction was better in large nodules (47.13 ± 21.51%) than in small nodules (30.25 ± 70.10%, *p* < 0.001) and medium nodules (40.85 ± 20.88%, *p* = 0.01) at the 1-month follow-up. These results are consistent with data from our previous findings [7], but are inconsistent with studies from other teams, which showed more effective reduction rates in small nodules [15,16]. The discrepancy of different studies may due to the ablation effect. We identified a mild increase in the size of small nodules observed on ultrasound immediately after ablation as the ablated range was extended beyond the nodular borders. At the 1-month follow-up, the increased size of small nodules would result in a lower VRR; the VRR was calculated by the initial nodular volume prior to RFA and the volume of the ablated region at the 1-, 3-, and 6-months after RFA. In the large nodular group, it was easier to identify the safety margin than in the small nodular group during ablation, thus limiting over-treated regions. The ablated large nodules did not show increased size immediately after RFA; therefore, the lower VRR of the small nodular group at the 1-month follow-up did not indicate a lower RFA efficacy. The three groups achieved similar VRR results at the 6-month follow-up (large group, 72.51% ± 18.52; medium group, 72.12% ± 18.23; small group, 74.62% ± 28.64, *p* = 0.28). Therefore, regardless of the size of solid and predominant solid thyroid nodules, optimal and acceptable VRR could be achieved by ablation.

In the present study, different specialists demonstrated variable VRR after 6-month follow-up (Appendix A). Until now, there was no related investigation focusing on this finding, and might be associated with lots of institutional issue in the beginning to carry out the similar procedure. Generally speaking, goiters with different sizes can attain a successful VRR (>50%) with all well-trained operators and department factors might be able to be neglected. More noteworthy, the average size of the initially selected BTNs was relatively smaller than in subsequent cases (volume of initial 50 cases: 19.01 ± 17.83 mL versus volume of the rest: 22.47 ± 29.83 mL, *p* = 0.087). After the initial RFA procedures, operators could have gained the confidence and skills necessary to ablate more complicated cases, and thereby achieve better outcomes (VRR of initial 50 cases: 67.64 ± 24.45% versus volume of the rest: 72.78 ± 17.77%, *p* = 0.002). It is reasonable to assume that thyroid RFA has a learning curve, whereby operators may achieve superior outcomes over time.

The overall cosmetic and symptomatic scores improved significantly after ablation, while interventional wounds and scarring also healed during the follow-up period. The primary factor causing the higher cosmetic and symptomatic scores among patients with larger nodules was higher initial nodular size prior to ablation. Accordingly, the scores of the large nodule group were higher than those of the small nodule group during follow-up. To achieve similar cosmetic and symptomatic scores, subsequent RFA procedures may be necessary in patients presenting with large nodules.

Of the 762 patients, 40 (4.8%) patients experienced complications. None of these complications were life threatening, or with sequelae related to RFA. Voice change was the most common complication during ablation, with most patients recovering their voice within 1 week to 6 months after the ablation procedure. The overall complication rate was significantly higher in the large nodule group than in the small and medium nodule groups (*p* < 0.001). Larger nodules demonstrated several characteristics which resulted in a higher complication rate, including closer proximity to the danger triangle or skin region, difficulty in determining the overall nodular margin, and post-ablation mass effect and increased intra-nodular pressure after RFA. The complication rate noted herein is similar to a previous multicenter study which reported a 3.3% complication rate [6]. During the establishment of RFA procedures for BTNs in Taiwan, a comprehensive understanding of potential complications and effective methodologies to prevent major complications was of critical importance. A recent review suggested that RFA is a safe alternative treatment for BTNs with minor complications [17].

There are several limitations to the present study. First, patients with a single nodule were enrolled for single-session ablation in this study. The efficacy of RFA for multiple thyroid nodules should be evaluated in another future study to confirm treatment outcomes noted in Taiwan. Another limitation was the lack of systematic follow-up. Retrospective analyses have a tendency to underestimate the true treatment outcomes as patients who present significantly improved cosmetic and symptomatic concerns may tend to cease follow-up. Third, the observation time was relatively short at 6 months. A future longitudinal follow-up study would be necessary to analyze and evaluate longer-term outcomes, including the risk and rate of regrowth.

In conclusion, this study demonstrates that single-session RFA is not only an effective treatment to reduce nodular volume, but also a safety treatment modality to improve cosmetic and symptomatic concerns. Ultrasound-guided RFA for benign thyroid nodules has evolved in a valuable alternative treatment in Taiwan. The various TATA training programs for different specialists have increased the accessibility to thyroid RFA, and are dedicated to progress and improvement in the field.

## Figures and Tables

**Figure 1 jpm-12-00063-f001:**
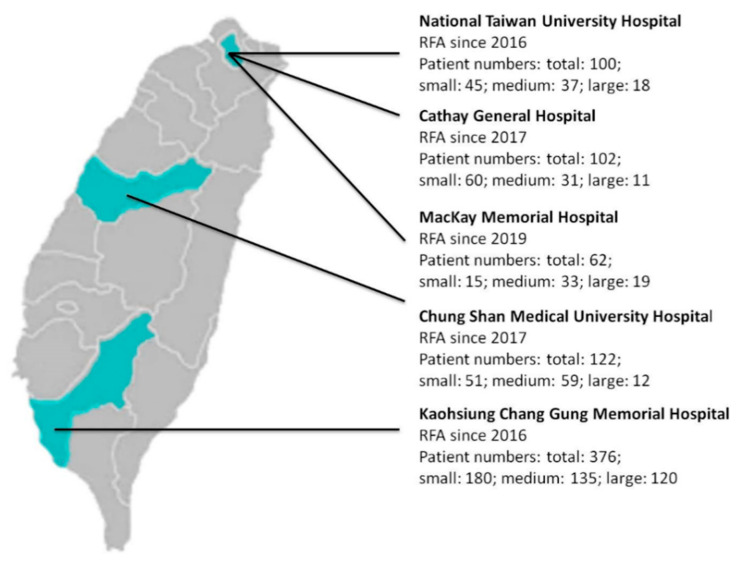
Participating medical centers.

**Figure 2 jpm-12-00063-f002:**
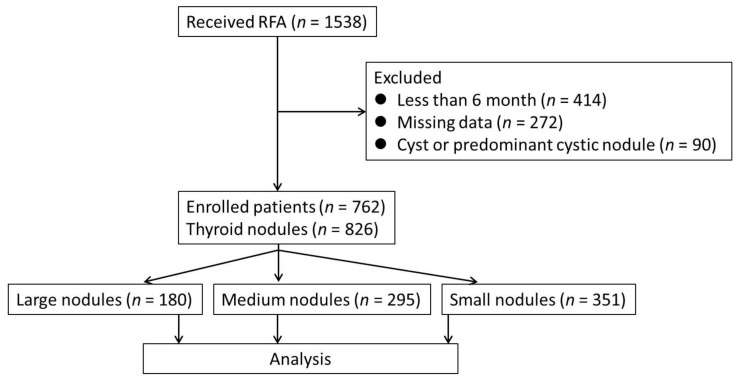
Consort diagram of the study.

**Figure 3 jpm-12-00063-f003:**
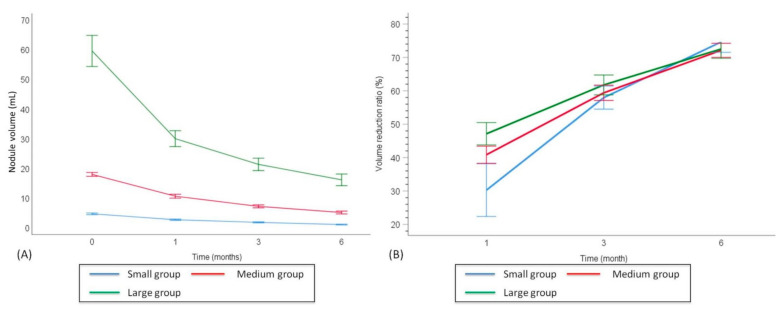
Follow-up changes in nodular volume and VRR of the three groups of BTNs. Changes in nodular volume (**A**) and VRR (**B**) of the different echogenicity groups of BTNs at each follow-up. BTN: benign thyroid nodule; VRR: volume reduction ratio.

**Table 1 jpm-12-00063-t001:** Demographic data based on nodule volume.

	Total	Small (≤10 mL)	Medium (11–30 mL)	Large (>30 mL)	*p*-Value
Number of patients	762	299	290	173	
Number of nodules	826	351	295	180	
Age Median	45.22 ± 11.97	44.44 ± 11.35	44.83 ± 12.60	47.40 ± 11.91	0.388
Sex, female/male	625 (82%)/137 (18%)	251/61	236/45	138/31	0.203
Laboratory studies (mean ± SD)					
Serum Free T4 (ng/dL)	1.17 ± 0.24	1.15 ± 0.20	1.19 ± 0.23	1.21 ± 0.31	0.140
Serum T3 (ng/dL)	105.58 ± 29.29	108.45 ± 30.65	101.80 ± 27.33	106.05 ± 29.20	0.045
Serum TSH (uIU/mL)	1.57 ± 1.19	1.79 ± 1.42	1.44 ± 1.01	1.37 ± 0.96	0.015
Technical characteristics					
Delivered energy (Kcal)	7.92 ± 7.30	2.64 ± 2.06	7.08 ± 4.68	14.14 ± 8.20	<0.001
Energy/mL ^a^ (Kcal/cm^3^)	0.51 ± 0.32	0.69 ± 0.53	0.38 ± 0.24	0.24 ± 0.12	<0.001
Ablation time (minutes)	17.73 ± 9.68	10.09 ± 5.19	17.88 ± 7.25	25.34 ± 9.28	<0.001

^a^ Energy/mL: energy delivered per milliliter of the pretreatment nodule volume. SD: standard deviation; ng: nanogram; dL: deciliter; uIU: mirco International Unit; kilocalorie; mL: milliliter; cm: centimeter.

**Table 2 jpm-12-00063-t002:** Outcomes based on nodule volume ^a^.

	Total (*n* = 826)	Small (≤10 mL, *n* = 351)	Medium (11~30 mL, *n* = 295)	Large (>30 mL, *n* = 180)	Group * Time Effect *p*-Value ^c^
Nodule volume (Mean ± SD, mL)					<0.001
Baseline	21.51 ± 27.13	4.77 ± 2.80	18.04 ± 5.65	59.60 ± 35.82	
1 month	11.71 ± 13.75	2.78 ± 2.02	10.70 ± 5.28	30.07 ± 17.29	
3 months	8.07 ± 10.23	1.89 ± 1.66	7.28 ± 4.15	21.42 ± 13.85	
6 months	5.94 ± 8.83	1.15 ± 1.28	5.20 ± 4.15	10.21 ± 13.18	
Time Effect *p*-value ^b^	<0.001	<0.001	<0.001	<0.001	
Volume reduction ratio (Mean ± SD, %)					<0.001
1 month	37.76 ± 48.80	30.25 ± 70.10 ^d,e^	40.85 ± 20.88 ^d,f^	47.13 ± 21.51 ^e,f^	
3 months	59.29 ± 25.65	57.95 ± 32.08 ^g,h^	59.39 ± 19.39 ^g,i^	61.76 ± 19.62 ^h,i^	
6 months	73.24 ± 23.19	74.62 ± 28.64 ^j,k^	72.12 ± 18.23 ^j,l^	72.51 ± 18.52 ^k,l^	
Time Effect *p*-value	<0.001	<0.001	<0.001	<0.001	
Cosmetic Score (0–3)					
Baseline	2.53 ± 1.09	2.01 ± 1.17	2.91 ± 0.89	3.00 ± 0.69	<0.001
6 months	0.91 ± 0.97	0.38 ± 0.62	1.11 ± 0.93	1.70 ± 0.98	<0.001
Time Effect *p*-value	<0.001	<0.001	<0.001	<0.001	
Symptoms Score (0–5)					
Baseline	1.75 ± 1.41	1.40 ± 1.33	2.02 ± 1.47	2.04 ± 1.33	<0.001
6 months	0.13 ± 0.42	0.03 ± 0.18	0.13 ± 0.39	0.34 ± 0.69	<0.001
Time Effect *p*-value	<0.001	<0.001	<0.001	<0.001	
Complications					
Voice change	22	10	4	8	
Delay nodular rupture	9	1	1	7	
Ptosis	2	1	0	1	
Hyperthyroidism	2	1	0	1	
Hematoma	4	0	1	3	
Skin burn	1	0	0	1	
Total	40 (4.8%)	13 (3.7%)	6 (2.0%)	21 (11.6%)	<0.001

^a^ Generalized estimation equation analysis was used for repeated measurement analysis of volume change and volume reduction ratio at different time outcomes. ^b^ Time effect defined as volume differences of the same group between different times. The baseline measurements were the reference categories. ^c^ Group * (multiplied by) time effect is defined as change of volume for different volume groups at 6 months. ^d^ *p*= 0.035; ^e^ *p* < 0.001; ^f^ *p* = 0.01; ^g^ *p* = 1.00; ^h^ *p* = 0.295; ^i^ *p* = 0.634; ^j^ *p* = 0.570; ^k^ *p* = 0.953; ^l^ *p* = 1.00.

## Data Availability

Data available upon request due to restrictions. The data presented in this study are available on request from the corresponding author.

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
