# Peer review of "Multicenter Study of Benign Thyroid Nodules with Radiofrequency Ablation: Results of 762 Cases over 4 Years in Taiwan"

_jpm, 2022, doi:10.3390/jpm12010063_

Round 1

Reviewer 1 Report

I find the manuscript titled “Multicenter study of benign thyroid nodules with radiofrequency ablation: Results of 700+ cases over 4 years in Taiwan” absolutely clear with very nice data results. My only suggestion is about the title. I was a little bit confused about 700+ cases. I find it more proper…. 762 cases over 4 years in Taiwan. Further, what are your suggestions about thyroid nodule pathogenesis in Taiwan, iodine deficiency? What is the thyroid nodule prevalence in Taiwan?

Author Response

1) I find the manuscript titled “Multicenter study of benign thyroid nodules with radiofrequency ablation: Results of 700+ cases over 4 years in Taiwan” absolutely clear with very nice data results. My only suggestion is about the title. I was a little bit confused about 700+ cases. I find it more proper…. 762 cases over 4 years in Taiwan.

Thank you for your suggestion. We will change the title more specific.

2) What are your suggestions about thyroid nodule pathogenesis in Taiwan, iodine deficiency? What is the thyroid nodule prevalence in Taiwan?

Mandatory salt iodization in Taiwan successfully reduced the goiter rate from 21.6% to 4.3% in school children surveyed in 1971. The adult goiter prevalence was found to be 19.4% in males, 33.6% in females and 25% in total. Nowadays, the most common cause of benign thyroid nodule in Taiwan is sporadic (multinodular goiter), followed by hereditary factors, environment factors, and autoimmune diseases.

Reference:

[1] Iodine status of adults in Taiwan 2005-2008, 5 years after the cessation of mandatory salt iodization. J Formos Med Assoc. 2016 Aug;115(8):645-51.

[2] Prevalence of goiter in Taiwanese adults: a preliminary study. J Formos Med Assoc 1995, 94:197-9.

[3] Etiology of adult goiter in Taiwan--a hospital-based study. Zhonghua Yi Xue Za Zhi (Taipei) 1991, 47:154-60.

Reviewer 2 Report

In the manuscript Lin et al. the authors present the result of a multicenter retrospective study aimed at evaluating the effectiveness and safety of radiofrequency ablation (RFA) treatment for benign thyroid nodules (BTNs) in on a total of 762 patients.

I have the following comments:

1) the manuscript needs a spell check: some parts, like the title could be more clear.

2) quality of figure 3 could be improved

3) the authors could be more specific on the technical characteristic of the treatment (Delivered energy, Energy/ml, Ablation time, and used electrode active tip (mm)).

4)  are there any information about the number of patients receiving treatment for other comorbidities?

Author Response

1) the manuscript needs a spell check: some parts, like the title could be more clear.

Thank you for your suggestion. We will change the title more specific and check the spelling and grammar of this article again.

2) quality of figure 3 could be improved

Thank you for your suggestion. We will insert figure with high resolution.

3) the authors could be more specific on the technical characteristic of the treatment (Delivered energy, Energy/ml, Ablation time, and used electrode active tip (mm)).

Thank you for your suggestion. We added more information about technical characteristic of the treatment, including delivered energy, Energy/ml, and ablation time, among three groups.

4)  are there any information about the number of patients receiving treatment for other comorbidities?

In this retrospective study, there are no complete data about comorbidities of subjects in 5 medical centers. In my institution, Kaohsiung Chang Gung Memorial Hospital, there are about 10% to 15% subjects with hypertension under medication treatment.